# Optimizing antibiotic stewardship and reducing antimicrobial resistance in Central Asia: A study protocol for evidence-based practice and policy

Lisa Lim[1]*, Laura Kassym[2], Assiya Kussainova[1], Bibigul Aubakirova[3], Yuliya Semenova[1]

1 Nazarbayev University, Astana, Kazakhstan, 2 Astana Medical University, Astana, Kazakhstan, 3 WHO Country Office in Kazakhstan, Astana, Kazakhstan

* lisa.lim@nu.edu.kz

**Data Availability Statement:** No datasets were generated or analyzed during the current study. All

## Abstract

Antimicrobial resistance (AMR) poses a significant global health challenge, prompting the World Health Organization (WHO) to stress the importance of monitoring antibiotic consumption and sales to address AMR effectively. This study protocol aims to optimize antibiotic stewardship and combat AMR in Central Asia through evidence-based practices and policies. The protocol includes objectives such as conducting systematic reviews of interventions to promote judicious antibiotic use, assessing antibiotic consumption trends, and investigating antibiotic overuse practices among healthcare providers. The study aims to raise awareness among stakeholders to enhance appropriate antibiotic prescribing practices. By establishing regulatory frameworks, monitoring policies' effectiveness, and providing training programs for healthcare professionals, this study seeks to contribute to the global efforts in combating AMR and promoting prudent antibiotic use.

## 1. Introduction

Antimicrobial resistance (AMR) is a growing global health issue, hence, the World Health Organization (WHO) emphasizes the need for countries to monitor antibiotic consumption and sales to combat AMR [1,2]. In 2019, nearly 4.95 million deaths were linked to bacterial AMR, with a range of 3.62 to 6.57 million. Among these deaths, 1.27 million were specifically attributed to bacterial AMR, with a range of 0.911 to 1.71 million [3]. By 2050, it is projected that AMR would be accountable for causing 10 million deaths annually on a global scale [4]. Thus, the presence of effective antimicrobial medications are necessary to combat the emergence and global spread of emerging infectious diseases. Despite the discovery of the most recent groups of antibiotics in 1984 [5], there remain multiple challenges in the process of introducing new antibiotics into the market. Ineffectiveness of existing available drugs in combating bacterial infections may result in higher morbidity rates, increased usage of healthcare resources, and an elevated risk of premature deaths [6]. Particularly with the aging population

relevant data from this study will be made available upon study completion.

**Funding:** This work was supported by Nazarbayev University under Collaborative Research Program Grant No. 211123CRP1609 awarded to YS (PI) and LL (co-PI).

**Competing interests:** The authors have decleared that no competing interests exist.

and the rising rates of obesity and diabetes, the importance of possessing effective antibiotics cannot be emphasized enough. Besides, the problem of AMR is complex, and no single public health campaign can address the whole spectrum of challenges associated with the spread of bacterial agents that are resistant to existing available antibiotics. Due to a variety of factors, including the availability of generics at low cost and the use of antibiotics as growth promoters, the past few decades have been characterized by the misuse of antibiotics in both animal and human medicine [7,8]. The issue of AMR is further compounded by unregulated sale of drugs without a proper prescription over-the-counter (OTC) and a lack of information regarding the recommended dosage limits [9,10]. Thereby, several international agencies have proposed a number of initiatives aimed at raising global awareness, increasing financial resources available for infectious diseases, and promoting a One Health approach that aims to balance the health of people, animals, and the environment [11,12].

AMR issues can be mitigated by reducing inappropriate use of antibiotics. The AMR issue and improper use of antibiotics, as well as limited understanding of implementation of antimicrobial stewardship (AMS) [13,14] efforts and factors that influence these practices in primary care settings, remain largely unknown in Kazakhstan and other Central Asian countries [15,16]. There have been separate studies evaluating the prevalence of antibiotic-resistant strains in hospital facilities [17,18], as well as attempts to assess antibiotic consumption at the inpatient level with data available up to 2019 [19–21]. Yet there is a lack of data for the post-COVID period, and considering only the inpatient healthcare sector may result in significant underestimation of actual antibiotic consumption. This is because the majority of patients receive treatment at the community level, and further the extent of self-medication is unclear. A recent local study of public awareness of antibiotic consumption and resistance among the adult population of Kazakhstan revealed that about 40% of respondents had received antibiotics without a prescription within the past year, and nearly 65% of them were unaware of AMR [22]. Nevertheless, since the study exclusively focused on the general people within the East Kazakhstan region, the applicability of these findings to the wider population of Kazakhstan remains uncertain.

Our proposed project is the first study to assess the factors influencing antibiotic prescription practices by a general practitioner [1] (GP), who plays a crucial role in primary healthcare settings in Kazakhstan and other Central Asian countries. Besides, this study attempts to explore the role of pharmacists in OTC sales of antibiotics and their approaches to manage leftover antibiotics. On the basis of the WHO Access, Watch and Reserve (AWaRe) Classification in 2019, the majority of antibiotics consumed at the level of hospital care in Kazakhstan and Central Asia belonged to the "Watch" group (68%), and this trend was growing similar to the consumption of antibiotics from the "Reserve" group [23]. We will apply AWaRe methodology to evaluate antibiotic consumption across Access, Watch, and Reserve categories at the community level in Central Asia. We will evaluate practices of non-prudent use of antibiotics in primary care settings, monitor AMS activities in primary care settings, and identify factors influencing their implementation in Kazakhstan and other Central Asian countries. Ultimately, our project findings will inform policy measures to improve antibiotic stewardship and reduce AMR in Kazakhstan and Central Asia.

## 1.1. Study objectives and anticipated outcomes

**1.1.1 Aims.** Our project aim is to conduct evidence synthesis of best practices to improve antibiotic stewardship and inform policy options to reduce antimicrobial resistance (AMR) in Central Asia. The sub-goals of the study are as follows:

- To investigate the drivers and determinants of non-prudent use and sales of antibiotics to develop targeted interventions at the community level to address AMR effectively in Central Asia.

- To assess the consumption of antibiotic drugs in defined daily doses (DDD) both in the community and hospital sectors, as well as estimate time trends in Central Asia.

- To generate in-depth knowledge of the reasons for over-the-counter (OTC) sales of antibiotics in the community pharmacies in order to evaluate the effectiveness of existing policies and regulations related to AMR control in Central Asian countries.

- To investigate antibiotic overuse by exploring antibiotic prescribing practices among primary care providers for common infections such as upper respiratory tract infections, otitis, and sinusitis in Central Asia.

- To conduct a comprehensive systematic review evaluating various interventions designed to promote judicious use of antibiotics among community members, general practitioners [24], and community pharmacists.

**1.1.2 Expected results.** The study aims to raise awareness among a wide range of stakeholders, including representatives of the Ministry of Health (MoH), National Health Insurance Fund (NHIF), academia, and the general public, regarding key factors associated with AMR and antibiotic use in Kazakhstan. The results of the study will:

- Establish a robust regulatory framework to conduct regular inspection of pharmacies, clinics, and other outlets where antibiotics are sold, including penalties for noncompliance with prescription requirements.

- Establish mechanisms to monitor the effectiveness of reinforcement policies and assess their impact on reducing OTC antibiotic sales.

- Provide comprehensive training programs and continuous medical education to healthcare professionals to enhance their knowledge of appropriate antibiotic prescribing practices.

- Develop and strengthen surveillance systems to monitor antibiotic sales and consumption patterns.

- Enhance the regulatory capacity for assessing the quality, safety, and efficacy of antibiotics available in the market.

- Develop and implement interventions aimed at reducing non-prudent antibiotic use and sale, such as educational campaigns, AMS programs, and interventions targeting specific healthcare sectors or patient populations.

To achieve this goal, we will convene a country dialogue meeting, disseminate the knowledge generated from our research, and build on research and training programs to enhance local expertise in AMR surveillance, laboratory diagnostics, infection prevention and control in Central Asia. These accomplishments will have important socioeconomic benefits by reducing overall antibiotic consumption, curbing OTC sales, and slowing the spread of antibiotic-resistant bacteria throughout the country.

## 2. Materials and methods

### 2.1 Study framework

The study procedure is depicted below in Fig 1, outlining the main areas of focus through a series of interconnected steps.

### 2.2 Data sources and methods for each target as follows

**Objective 1**: To investigate the drivers and determinants of non-prudent use and sale of antibiotics at the community level to develop targeted interventions to address the antimicrobial resistance (AMR) issues in Central Asia. Three related sub-studies will be implemented to identify the primary factors of antibiotic misuse among (i) adult residents, (ii) general practitioners, (iii) prospective medical doctors at the community level.

*Sub-study 1 for Objective 1*: *To identify the key factors contributing to the non-prudent use of antibiotics in adult populations at the community level.* This study involves community residents in both sexes aged 18 years and older. The survey will utilize the questionnaire presented in the WHO report "*Antibiotic Resistance*: *Multi-country Public Awareness Survey*" [25]. The questionnaire was previously piloted in a cohort of adult individuals aged 18 years and older from the East Kazakhstan province [22]. The calculations for the sample size will be conducted using the following methodology, with Kazakhstan serving as an illustrative example. The survey sample will comprise 683 individuals based on the calculation made with the help of EpiInfo7. The sample size calculation was based on the approximately 13,300,000 adult populations in Kazakhstan, an expected frequency of good knowledge of AMR (around 20%) [22], an acceptable margin of error (3%), a 95% confidence interval, and a design effect of 1.0. The sampling design will have two domains. The first domain will include all settlements except the largest cities, and the second domain will consist only of these largest cities. Four sampling

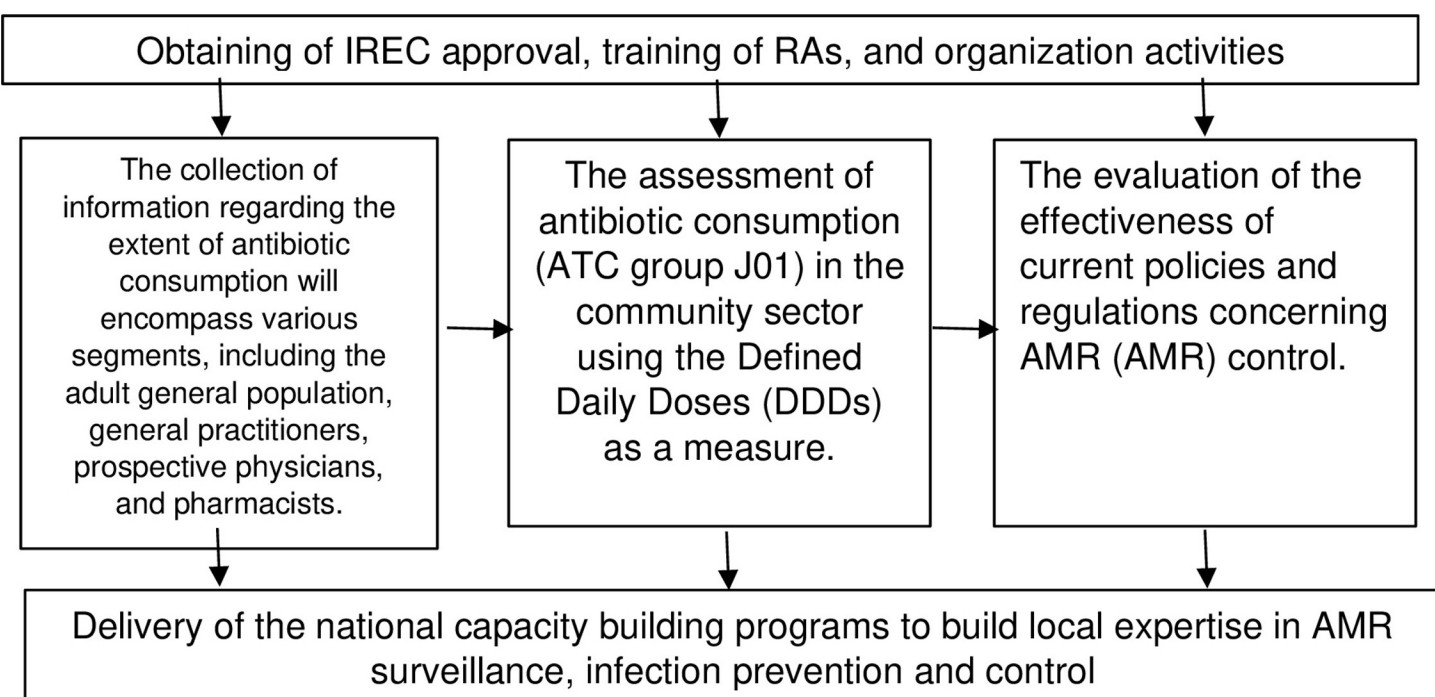

**Fig 1. Illustration of primary areas in connected phases.**

stages will be used in the first domain. At the first stage, settlements will be selected using stratified sampling (SS), with selection probability based on population size. Stratification will be done by five geographic zones–central, north, east, south, and west. From each selected settlement, two electoral districts will be selected using simple random sampling (SRS). Dwelling units in each electoral district will be selected by SRS, with a total of 18 units per district. From each dwelling unit, one eligible person will be selected using the Kish Grid. The second domain will involve three sampling stages. At the first stage, a SS of electoral districts will be selected, with stratification based on the three largest cities. The second and third stages (sampling of dwelling units and individuals) will be identical to the first domain. Interviews will be conducted face-to-face in local languages, depending on the respondent's choice. The study objectives will be clearly explained and communicated before the interview starts, and informed consent will be obtained.

***Sub-study 2 for Objective 1***: *To evaluate the knowledge, attitudes, and behaviors of general practitioners with respect to antibiotics, antibiotic use, and antibiotic resistance.* The survey will be used to evaluate the knowledge, attitudes and behaviors of general practitioners regarding antibiotics and AMR by using the questionnaire developed by the European Centre for Disease Prevention and Control (ECDC) [25]. The questionnaire was previously piloted in a cohort of GPs from Astana. The calculations for the sample size will be conducted using the following methodology, with Kazakhstan serving as an illustrative example. According to calculations made with the help of EpiInfo7, it has to include a minimum of 690 individuals in order for the study to be generalizable. This calculation is based on the number of general 5,000 practitioners in Kazakhstan, an expected frequency of an adequate knowledge around 75% of AMR [26], with an acceptable margin of 3% error, a 95% confidence interval, and a design effect of 1.0. The sampling frame for this study will be composed of members of the Kazakhstan Association of Family Physicians (KAFP). To select the study participants, 750 physicians will be randomly sampled from the sampling frame, and an invitation to participate in an online survey will be sent to them. The invitation will explain the survey aim, provide a declaration of informed consent, and offer a link to the questionnaires. The online questionnaires will be open for responses over a 6-week period, and will be available in both Russian and Kazakh languages. The same sampling size and frame will be used in other Central Asian countries.

***Sub-study 3 for Objective 1***: *To evaluate the knowledge, attitudes, and behaviors of prospective medical doctors with respect to antibiotics, antibiotic use and antibiotic resistance.* This study will involve final year medical students from all six medical schools in Kazakhstan. The survey will utilize a questionnaire developed by the ECDC, which includes a separate subsection for health students that focuses on the AMR training received in medical schools [26]. In addition to this subsection, the students will respond to questions from other subsections of the questionnaire, except for those related to prescribing practices. The calculations for the sample size will be conducted using the following methodology, with Kazakhstan serving as an illustrative example. In order for the study to be generalizable, it must include a minimum of 723 participants, which has been calculated using *EpiInfo7*. This calculation is based on the total number of final year medical students in Kazakhstan (7,500), an expected frequency of adequate knowledge of approximately 75% AMR [26], an acceptable margin of 3% error, a 95% confidence interval, and a design effect of 1.0. The final year medical students will be selected randomly, based on the lists provided by the deans of their respective medical schools. An invitation to participate in an online survey will be sent to 800 final year medical students. The invitation will provide details about the survey's purpose, include a declaration of informed consent, and offer a link to the questionnaire. The online questionnaire will be available for responses over a 6-week period and will be provided in both Russian and Kazakh languages. The same sampling size and frame will be carried out in other Central Asian countries.

Statistical analysis will be conducted using IBM Statistical Package for Social Sciences (SPSS) version 24 software and Microsoft Excel. Continuous variables will be presented as mean and standard deviation or median and interquartile range, depending on the data distribution, which will be evaluated by the Kolmogorov-Smirnov test and histograms. Categorical variables will be presented as absolute numbers and proportions. Pearson's chi-square test will be used for between-group comparisons of categorical variables, while Student's t-test/ANOVA or Mann-Whitney/Kruskal-Wallis test will be used for normally and non-normally distributed continuous variables, respectively. Five-point agreement responses will be collapsed into two categories: (i) agree, undecided/neither agree, nor disagree, disagree and (ii) once a day/more than once a day, once a week/more than once a week, and rarely/never. Logistic regression will be conducted to identify factors associated with non-prudent use of antibiotics.

**Objective 2**: To assess the consumption of antibacterial drugs in defined daily doses (DDD) in both the community and hospital sectors, as well as estimate time trends. The research team will conduct two distinct sub-studies to achieve the objective: (i) by assessing antibiotic consumption in the community and hospital sectors, and (ii) conducting a time-series analysis of country-level antibiotic consumption over a 10-year period.

***Sub-study 1 for Objective 2***: *To assess antibiotic consumption in the community sector*. A pharmaceutical consulting company "VIORTIS" collects data on sales of all medications in all pharmacies, as well as the data on the medicines supplied to hospital facilities. Therefore, this data can be used to evaluate the consumption of antibiotics both at the community and hospital sectors with high precision. The consumption of antibiotics (ATC group J01) will be expressed in DDD/1000 inhabitants/day (DID) following the ATC/DDD (Anatomical Therapeutic Chemical/Defined Daily Dose) methodology developed by the WHO [27]. The demographic data for the relevant years will be collected from the statistical compilations issued by the Bureau of National Statistics [28]. The AWaRe methodology [29] will be used to evaluate the distribution of antibiotics consumed within the three groups: Access, Watch, and Reserve.

***Sub-study 2 for Objective 2***: *Conducting a time-series analysis of country-level antibiotic consumption over a 10- year period*. The study will analyze 10 years' worth of data (2015–2024) on country-level antibiotic consumption, disaggregated by the Access, Watch, and Reserve groups. Special attention will be given to antibiotic consumption during the COVID-19 pandemic when comparing with pre-pandemic and post pandemic periods. Statistical analysis will be conducted using the "Forecasting" package of IBM SPSS version 24 software and Microsoft Excel.

**Objective 3**: To generate in-depth knowledge of the reasons for OTC sales of antibiotics in the community in order to evaluate the effectiveness of existing policies and regulations related to AMR control in Central Asian countries. This objective aims to carry out three distinct sub-studies: (i) to monitor the practice of OTC antibiotic sales with different stakeholders in the community; (ii) to evaluate the effectiveness of reinforcement policies and assess their impact on reducing OTC antibiotic sales; and (iii) to conduct a desk review on the existing AMR policies, guidelines, and legislations, including implementation and assessment of national action plans (NAPs) on AMR.

***Sub-study 1 for Objective 3***: *To investigate the practice of OTC antibiotic sales from the perspective of the community pharmacists*. This qualitative study will employ in-depth interviews with community pharmacists working in metropolis, city, and rural areas. The sample for this study will be selected through convenience sampling, based on an equal distribution of urban and rural pharmacists, and their willingness to participate. Triangulation will be involved in utilizing different data collection order to check the consistency of the findings [30]. The investigation will utilize a flexible topic guide covering critical areas such as (i) the practice of and reasons for OTC antibiotic sales; (ii) knowledge and understanding of AMR and related antibiotic consumptions; (iii) awareness of existing legislation regarding OTC antibiotic sales; and

(iv) the existence of legal regulations promoting the accurate dispensing of prescribed antibiotic pharmaceutical forms.

***Sub-study 2 for Objective 3***: *To explore the mechanisms to evaluate the effectiveness of reinforcement policies and assess their impact on reducing OTC antibiotic sales*. This study will be a qualitative study conducted through in-depth interviews with key stakeholders, including representatives from the MoH, the NHIF, professional associations, and public health experts. Our investigation will utilize a flexible topic guide to cover critical areas such as (i) NAPs on AMR and antibiotics; (ii) national laws or regulations aimed at reducing the non-prudent use of antibiotics, including surveillance and auditing, including penalties for non-compliance with prescription requirements and other relevant regulations; (iii) legal regulations encouraging the accurate dispensing of prescribed antibiotic pharmaceutical forms, prosecution of illegal means of obtaining antibiotics. Prior to the interview, all participants will provide oral informed consent, which will include an agreement for audio recording. The interviews will be conducted in local languages, depending on the interviewee's preference. Recorded interviews will be transcribed verbatim and translated into English prior to coding. The transcribed text will be reviewed several times for familiarization. The inductive thematic analysis will be performed concurrently with data collection. A full and inclusive description of themes will be developed, and feedback from the participants will be sought to obtain verification of the authors' understanding of data. Statistical analysis will be conducted using the Version 14 of the NVivo software.

***Sub-study 3 for Objective 3***: *To conduct the desk literature review on the existing AMR policies, guidelines, and legislations, including implementation and assessment of the NAPs on AMR*. The present study will involve a desk review of existing laws, rules, and regulations. The review will be conducted using the Google Search engine, employing the following search queries: (i) "antimicrobials"; (ii) "AMR"; (iii) "OTC sale"; and (iv) "measures." Additionally, the national legislative databases managed by the Ministry of Justice will be accessed. The titles of all relevant documents will be evaluated, and their complete texts will be obtained and thoroughly examined. Data extraction and organization will be performed, utilizing tables specifically designed for this purpose.

**Objective 4**: To investigate antibiotic overuse by exploring antibiotic prescribing practices for common infections such as upper respiratory tract infections, otitis, and sinusitis in Central Asia among primary care providers. The provision of clinical care in Kazakhstan and other Central Asian countries is regulated by national standards of care, which are mandatory for all healthcare providers, including those in the private sector. These standards, referred to locally as "clinical protocols," guide the management of various medical conditions. In the first phase of this study, we will create a comprehensive list of available clinical protocols for treating upper respiratory tract infections. In the second phase, we will extract provisions concerning the use of antibiotics and compare them with current international evidence-based clinical practice guidelines developed by established professional associations in the field. We will organize data on the indications for antibiotic prescription, type of antibiotic, recommended duration of treatment, and dosage into tables designed for this purpose. This will enable us to identify areas for improvement in aligning current standards of care with evidence based international practices.

**Objective 5**: To conduct a comprehensive systematic review of systematic reviews evaluating various interventions designed to promote judicious use of antibiotics among two key groups: (i) community members and (ii) General Practitioners (GPs) and community pharmacists. To achieve this objective, three academic databases, namely PubMed, Google Scholar, and Science Direct, will be searched without any restrictions on publication year. This approach is considered suitable as it provides a comprehensive overview of interventions and

prevents the inclusion of redundant studies. In the initial step, two researchers will independently screen article titles and abstracts, and any discrepancies will be resolved by discussion with a third researcher. The study's data will include the first author, year of review publication, types of interventions assessed, their effectiveness (if available), number of primary studies included in each review, time period, world region, and main conclusions. A quality assessment tool will be employed to evaluate the methodological quality of the systematic reviews included.

## 2.3. Ethical consideration

This research will involve human subjects, and therefore all documents related to the study, including the application to the Nazarbayev University (NU) Institutional Research Ethics committee (IREC), informed consent forms, and questionnaires, will be submitted for approval by NU IREC. Data collection will only begin after approval has been granted. Informed consent will be obtained from each subject prior to data collection, and all documents as well as study results will be kept confidential. This study will not entail any physical activities or medical interventions, and will not address sensitive or personal topics. Data collection for Objectives 1, 2, and 4 will be anonymized, with no identifying information collected. Researchers responsible for the statistical analysis for Objectives 1–3 will be blinded. The research team will follow the principles of research ethics, avoiding scientific data fabrication, scientific fraud, plagiarism, false co-authorship, and scientific misconduct. Only questionnaires developed and made available for the general public for free by the WHO, WOAH, ECDC, and the European Commission will be used, and proper credit will be given to the developers upon utilization of these questionnaires. The recruitment period of this study will be August 2024-March 2025.

**Institutional Review Board Statement (IREC)**: This study is being conducted in accordance with the Declaration of Helsinki. The approval for Objective 2 (To assess the consumption of antibacterial drugs in defined daily doses (DDD) in both the community and hospital sectors, as well as estimate time trends) was taken from the Nazarbayev University (NU) IREC (#802/23112023, December 1, 2023). In addition, the necessary authorizations to collect survey and interview data for other objectives were received from the NU IREC (survey data collection- #851/05022024) and interview data collection- #861/20022024).

## 3. Discussion

### 3.1. The potential social and economic impact of the project for Central Asia

According to the WHO, antimicrobial resistance (AMR) constitutes one of the three greatest threats to human health in the coming decades [31]. The potential costs associated with the improper use of antibiotics are challenging to quantify. Estimates suggest that in the United States (USA) alone, the annual costs associated with AMR amount to 55 billion USD, with 20 billion USD attributed to direct healthcare expenses and approximately 35 billion USD representing productivity losses [23]. In the European Union (EU), AMR is responsible for 33,000 deaths per year and costs 1.5 billion EUR in additional healthcare expenses and productivity losses [31]. The O'Neill report projects that by 2050, drug-resistant diseases could cause 10 million deaths annually, equivalent to the economic impact of the 2008–2009 global financial crisis [3]. Furthermore, as many as 24 million people could be pushed into extreme poverty by 2030 [4]. We must address how AMR also impacts on environment, social, and economic

targets in the Sustainable Development Goals (SDG) framework, including SDG 3 –good health and wellbeing of all ages [32].

Recently, the issue of AMR has received significant attention from the medical community in Kazakhstan. Kazakhstan has developed a national action plan (NAP) on AMR [33]. The NPA primarily focuses on AMR surveillance, but inadequate attention is paid to controlling antibiotic use. Therefore, urgent steps are required to prioritize the funding, implementation, and monitoring interventions aimed at promoting prudent use of antibiotics. It is crucial to coordinate efforts across sectors and engage a diverse range of stakeholders, as this is likely to yield favorable outcomes. However, no attempt was made to estimate the social and economic consequences of AMR in Kazakhstan, even though they are likely to be significant. The economic costs include loss of productivity and expenditures on additional diagnostics and treatments, while social impact includes decreased quality of life, an upsurge in morbidity and mortality rates, amplified poverty rates, and exacerbated gender inequities [34]. Despite the affordability of antibiotics for the majority of the population in Kazakhstan and other Central Asia nations, coupled with a low knowledge and awareness of the rationale for consumption of antibiotics, this issue is likely to worsen without interventions and policies. Furthermore, the period of the COVID-19 pandemic has been characterized by increased consumption of medicines, including antibiotics, and high rates of self-medication [35], suggesting that the magnitude of the problem may have increased even further.

## Supporting information

**S1 Fig. Data management plan.**
(PDF)

**S2 Fig. Study management plan.**
(PDF)

**S3 Fig. Team composition.**
(PDF)

**S4 Fig. Research alignment.**
(PDF)

## Acknowledgments

I express my gratitude for the helpful discussions received from Nazarbayev University Graduate School of Public Policy and School of Medicine. I would also like to express my gratitude for the helpful discussions provided by the Department of Antimicrobial Resistance at the International Vaccine Institute.

## Author Contributions

**Conceptualization:** Lisa Lim, Yuliya Semenova.

**Funding acquisition:** Lisa Lim, Yuliya Semenova.

**Methodology:** Lisa Lim, Laura Kassym, Assiya Kussainova, Bibigul Aubakirova, Yuliya Semenova.

**Project administration:** Yuliya Semenova.

**Supervision:** Lisa Lim.

**Validation:** Lisa Lim.

**Writing – original draft:** Lisa Lim, Yuliya Semenova.

**Writing – review & editing:** Lisa Lim, Laura Kassym, Assiya Kussainova, Bibigul Aubakirova, Yuliya Semenova.

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
