## [Decision Letter · Decision Letter 0]

22 Oct 2024

PONE-D-24-28402Optimizing Antibiotic Stewardship and Reducing Antimicrobial Resistance in Central Asia:

A Study Protocol for Evidence-Based Practice and PolicyPLOS ONE

Dear Dr. Lim,

Thank you for submitting your manuscript to PLOS ONE. After careful consideration, we feel that it has merit but does not fully meet PLOS ONE’s publication criteria as it currently stands. Therefore, we invite you to submit a revised version of the manuscript that addresses the points raised during the review process.

We look forward to receiving your revised manuscript.

Kind regards,

Arianit Jakupi, PhD

Academic Editor

PLOS ONE

Journal Requirements:

2. Thank you for stating the following in your manuscript:

“This research is funded by the Nazarbayev University under the Collaborative Research Project (CRP) Grant № OPCRP2024008 of “Evidence-based practice and policy to improve antibiotic stewardship and reduce antimicrobial resistance in Central Asia”

 ” Grant program: Collaborative Research Project 2024-2026

Funder-Project Reference: 211123CRP1609”

4. Please amend the manuscript submission data (via Edit Submission) to include authors Laura Kassym , Assiya Kussainova , Yuliya Semenova , and Bibigul Aubakirova.

6 Please include captions for your Supporting Information files at the end of your manuscript, and update any in-text citations to match accordingly. Please see our Supporting Information guidelines for more information: http://journals.plos.org/plosone/s/supporting-information.

Reviewers' comments:

Reviewer's Responses to Questions

**Comments to the Author**

1. Does the manuscript provide a valid rationale for the proposed study, with clearly identified and justified research questions?

Reviewer #1: Partly

2. Is the protocol technically sound and planned in a manner that will lead to a meaningful outcome and allow testing the stated hypotheses?

Reviewer #1: Partly

3. Is the methodology feasible and described in sufficient detail to allow the work to be replicable?

Reviewer #1: Yes

4. Have the authors described where all data underlying the findings will be made available when the study is complete?

Reviewer #1: Yes

5. Is the manuscript presented in an intelligible fashion and written in standard English?

Reviewer #1: Yes

6. Review Comments to the Author

You may also provide optional suggestions and comments to authors that they might find helpful in planning their study.

Reviewer #1: The abstract you've written is clear and focused. The first sentence can be made more impactful by specifying why AMR is a global challenge. You mention the establishment of regulatory frameworks but don't detail the nature of these frameworks. Consider providing more specifics in the full text (not the abstract), but it’s good to leave a hint here. There are some areas where phrasing and structure could be slightly improved for a smoother read and alignment with academic expectations. Abstracts should not include abbreviations, if possible. Abbreviation Consistency: Ensure all abbreviations like AMS, DDD, and OTC are introduced at first mention and used consistently throughout the paper.

You introduce several study goals at once; it may help to break them down slightly for clearer readability. Consider clarifying who the stakeholders are. Are they healthcare providers, policymakers, or the general public? You mention the establishment of regulatory frameworks but don't detail the nature of these frameworks. Consider providing more specifics in the full text

Methodological Section: Consider reducing long sentences for ease of understanding. Certain sections would benefit from more specificity, particularly in sampling procedures and desk review methodologies. Additionally, clarity on how results will translate into policy recommendations or interventions would improve the impact and practical application of the findings.

Ethical Consideration: The ethical considerations outlined are appropriately thorough and reflect a strong commitment to maintaining high ethical standards. One suggestion would be to include a plan for how the anonymized data will be stored and managed post-study to ensure compliance with data protection regulations. Clarifying this, along with the process of obtaining informed consent would strengthen the ethical framework.

Discussion: The paper provides a solid foundation for understanding the urgency of addressing AMR in Central Asia, with well-structured objectives. However, more clarity on how the study findings will translate into policy recommendations or real-world interventions would enhance the impact of the research. The discussion could also benefit from a more comprehensive exploration of potential economic and social consequences specific to the region.

Take into consideration separating these two sections clearly (Discussion and Conclusion). This will allow the discussion to focus on analysis and exploration, while the conclusion to give final summary and closes the argument.

7. PLOS authors have the option to publish the peer review history of their article (what does this mean?). If published, this will include your full peer review and any attached files.

Reviewer #1: No

---

## [Author Response · Author response to Decision Letter 0]

31 Oct 2024

RESPONSE TO REVIEWERS 

Reviewer’s comment 1

Response to comment 1: 

The manuscript has been revised to comply with PLOS ONE’s style requirements, including appropriate file naming conventions. 

• Page and line numbers were added, 

• Used double-spaced, Removed Footnotes, 

• Used “Vancouver” style for reference style

• Changed Level 1 Heading to Bold type, 18pt font 

• Changed Level 2 Heading to Bold type, 16pt font

• Changed Level 3 Heading to Bold type, 14pt, font

Reviewer’s comment 2

Thank you for stating the following in your manuscript:

“This research is funded by the Nazarbayev University under the Collaborative Research Project (CRP) Grant № OPCRP2024008 of “Evidence-based practice and policy to improve antibiotic stewardship and reduce antimicrobial resistance in Central Asia”

 ” Grant program: Collaborative Research Project 2024-2026 Funder-Project Reference: 211123CRP1609”. 

Response to comment 2: 

All funding-related text has been entirely removed from the manuscript. 

Reviewer’s comment 3. 

When completing the data availability statement of the submission form, you indicated that you will make your data available on acceptance. We strongly recommend all authors decide on a data sharing plan before acceptance, as the process can be lengthy and hold up publication timelines. Please note that, though access restrictions are acceptable now, your entire data will need to be made freely accessible if your manuscript is accepted for publication. This policy applies to all data except where public deposition would breach compliance with the protocol approved by your research ethics board. If you are unable to adhere to our open data policy, please kindly revise your statement to explain your reasoning and we will seek the editor's input on an exemption. Please be assured that, once you have provided your new statement, the assessment of your exemption will not hold up the peer review process.

Response to comment 3: 

We have included the data availability declaration. 

Line 386-387

Data Availability Statement

The data used and analysed during the current study are available from the corresponding author upon reasonable request.

Reviewer’s comment 4

Please amend the manuscript submission data (via Edit Submission) to include authors Laura Kassym , Assiya Kussainova , Yuliya Semenova , and Bibigul Aubakirova.

Response to comment 4:

We have included the following statement regarding the data agreement below. 

Line 21- 22

All authors have agreed to provide the data following the acceptance of the manuscript for publication.

Reviewer’s comment 5

Your ethics statement should only appear in the Methods section of your manuscript. If your ethics statement is written in any section besides the Methods, please delete it from any other section.

Response to comment 5: 

Ethics statement appears in the Methods section of the manuscript only.

Line 317-338. 

2.3 Ethical consideration

 Instructional Review Board Statement (IREC)

Reviewer’s comment 6 

Please include captions for your Supporting Information files at the end of your manuscript, and update any in-text citations to match accordingly. Please see our Supporting Information guidelines for more information: http://journals.plos.org/plosone/s/supporting-information.

Response to comment 6:

We've already incorporated captions for our supporting information in the manuscript accordingly, and ensured that all in-text citations match these captions.

Line 489-501

Supporting information

S1 Fig. This is the S1 Fig Title. This is the S1 Fig legend.

S1 Text. Data management plan.

S2 Fig. This is the S2 Fig Title. This is the S2 Fig legend.

S2 Text. Study management plan.

S3 Fig. This is the S3 Fig Title. This is the S3 Fig legend.

S3 Text. Team composition.

S4 Fig. This is the S4 Fig Title. This is the S4 Fig legend.

S4 Text. Research alignment.

Reviewer’s comment 7

Response to comment 7:

 We have reviewed all reference list to ensure that it is complete and correct.

---

## [Editor Report · Decision Letter 1]

6 Nov 2024

Optimizing Antibiotic Stewardship and Reducing Antimicrobial Resistance in Central Asia:

A Study Protocol for Evidence-Based Practice and Policy

PONE-D-24-28402R1

Dear Dr. Lim,

We’re pleased to inform you that your manuscript has been judged scientifically suitable for publication and will be formally accepted for publication once it meets all outstanding technical requirements.

Kind regards,

Arianit Jakupi, PhD

Academic Editor

PLOS ONE

---

## [Editor Report · Acceptance letter]

3 Jan 2025

PONE-D-24-28402R1 

PLOS ONE

Dear Dr. Lim, 

I'm pleased to inform you that your manuscript has been deemed suitable for publication in PLOS ONE. Congratulations! Your manuscript is now being handed over to our production team.

Kind regards, 

on behalf of

Dr Arianit Jakupi 

Academic Editor

PLOS ONE